# Intrinsic and extrinsic factors associated with sputum characteristics of presumed tuberculosis patients

**Fred Orina[1]\*, Moses Mwangi[2], Hellen Meme[1], Benson Kitole[3], Evans Amukoye[1]**

**1** Center for Respiratory Diseases Research, Kenya Medical Research Institute, Nairobi, Republic of Kenya,
**2** Center for Public Health Research, Kenya Medical Research Institute, Nairobi, Republic of Kenya,
**3** Malindi Sub-County Hospital, KIlifi, Republic of Kenya

☯ These authors contributed equally to this work.
\* orinafred@gmail.com

**Data Availability Statement:** All relevant data are within the manuscript and its Supporting Information files.

## Abstract

### Background

Sputum remains the most preferred specimen for detection of *Mycobacterium tuberculosis* due to its non-invasive method of production. Good quality sputum specimen is essential for accurate diagnosis of pulmonary tuberculosis (PTB). It is therefore imperative to assess factors that are related to the production of sputum that is of the best quality.

### Objective

We assessed the intrinsic and extrinsic characteristics of presumed tuberculosis patients and the quality of sputum they produced.

### Methods

This was a cross-sectional study in which consenting enrolled presumed tuberculosis patients were subjected to medical examination and a structured questionnaire administered to collect clinical history, demographic information, environmental and behavioral characteristics. The enrolled participants were instructed on how to collect spot and morning sputum specimens for macroscopic and microscopic assessment to determine any association.

### Results

A total of 309 patients were enrolled into the study with an even distribution on gender (50.5% males). Of these, 202 (65.3%) submitted both a spot and a morning specimen for analysis. On macroscopic examination, 70% spot and 68% morning sputum were characterized as good quality (Purulent/mucoid). The factors associated (p<0.05) with quality specimen included both intrinsic and extrinsic factors. The intrinsic factors included: difficulty in breathing, presence of conjunctivitis and knowledge of the disease whereas the only extrinsic factor associated with production of good quality sputum for tuberculosis diagnosis was time taken by patient to seek tuberculosis treatment after occurrence of any of the TB symptoms.

**Funding:** This work was supported by the Kenya Medical Research Institute, Internal Research Grant L308 to FO (www.kemri.org). Total funding received was $8000. The funders had no role in study design, data collection and analysis, decision to publish, or preparation of the manuscript.

**Competing interests:** The authors have declared that no competing interests exist.

## Conclusion

Both intrinsic and extrinsic factors affected the quality of sputum produced by presumed tuberculosis patients. Clinical and behavioral characteristics including conjunctivitis, difficulty in breathing and delay in seeking treatment were important factors that determined the production of good quality sputum specimens, while knowledge of tuberculosis disease did not compel presumed tuberculosis patients to produce good quality sputum for diagnosis of the disease.

## Introduction

Tuberculosis (TB) continues to be a global public health challenge with infections exceeding the human immunodeficiency virus (HIV) and malaria and is considered the largest cause of death from a single infectious disease. The World Health Organization (WHO) estimates that there are 10.4 million new cases and 1.7 million deaths annually [1]. The problem is further aggravated by the co-infections with HIV, diabetes and an increasingly aging population as well wider use of concomitant medication. Progression risk from latent to active TB is estimated to be between 16 and 27 times greater in people living with HIV than among those without HIV infection and Sub-Saharan Africa bears the brunt of the dual epidemic, accounting for approximately 86% of all deaths from HIV-associated TB in 2016 [2] The transmission of TB primarily depends on exogenous factors and is determined by an intrinsic combination of the infectiousness of the source case, proximity to contact and social and behavioural risk factors including smoking, alcohol, and indoor air pollution [3]. However, information on composition and the impact of changes in the oral–nasal cavity and lung microbiota on *M. tuberculosis* and how it establishes infection in the lower respiratory tract is limited [4]. This may contribute to variability of clinical manifestation of the active disease especially with mild or extensive pulmonary involvement, extra-pulmonary, or disseminated forms of TB [5].

To curb transmission of more infections, uncovering the links between endo, exo-environments of host and the pathogen, in particular understanding the sputum related characteristics with TB infection or disease is imperative. Immunologically, the airway mucosa responds to infection and inflammation in a variety of ways. These responses often include goblet cell and sub-mucosal gland hyperplasia and hypertrophy, with mucus hyper-secretion. Products of inflammation, including neutrophils, effete cells, bacteria, and cell debris, all contribute to mucus purulence and, when this is expectorated it is called sputum [6]. Expectorated sputum, the biological material for detection of pulmonary tuberculosis is mucous material from the lungs that is produced through coughing. It is most preferred for diagnosis due to its non-invasive method of collection. Sputum samples containing very little saliva is considered the best in TB diagnosis [7,8]. Purulent sputum has always been considered the best especially for culture and is characterized by elevated levels of lipid, DNA, and non-mucin proteins. However, Mucus glycoprotein form the basic biochemical constituent of sputum [9]. The production of purulent sputum is common also in patients with COPD conditions such as cystic fibrosis patients. Poor sputum specimens have basically been considered to be salivary. To reduce delays in diagnosis of tuberculosis the quality of the specimens must be optimum [10]. This may be an important issue especially in people living with HIV who have smear-negative pulmonary and extra-pulmonary tuberculosis.

Even with the deployed new diagnostic tools, their effectiveness may not be if the quality of specimen is not satisfactory. Determination of factors that affect the quality of sputum produced by tuberculosis patients, and specifically that influence the production of good quality sputum may be important in deducing the alternative methods in patient diagnosis or in

developing mechanisms that encourage proper sputum collection for accurate pulmonary tuberculosis diagnosis thereby ensuring early diagnosis and subsequent reduction of disease transmission within the population.

## Methods

### Study design

The cross-sectional study was conducted from January 2017 to October 2018 in a single referral hospital in the coastal region of Kenya. Presumed pulmonary tuberculosis participants were enrolled consecutively to assess factors that were associated with the sputum specimen they produced.

### Administration of study tools

Two data collection tools were used for collection of patient information. (1) A pretested self-administered questionnaire for collection of bio-data, knowledge, attitude practice, comorbidity and use of concomitant medication on the enrolled patients. (2) A standardized clinical form was for collection of medical information including parameters on clinical symptoms, general condition of the patient, their cardiovascular and respiratory systems among other significant symptoms.

### Sputum collection

All study participants received the sputum collection instructions including; collection environment, recommended posture, the procedure promoting deep coughing to obtain a sputum specimen from the lower lung, the approximate amount of time to be taken for correct volume to be acquired, and specimen handling procedure after collection. Participants were requested to collect two sputum specimens i.e. spot and morning samples. The spot specimen was collected at recruitment when the patient attended the clinic for the first time while the morning one was collected as early morning sample at home the following day. The specimens were then characterized macroscopically and microscopically by skilled retrained laboratory staff.

### Macroscopic characterization of sputum

This was done as previously described by Yoon and colleague collogues [11]. Sputum that was clear and watery appearance without any viscosity was categorized as saliva. The differentiation between mucoid and purulent sputa was based on a five-point sputum color chart (BronkoTest; Heredilab Inc., Salt Lake City, UT, USA). Colors 1 and 2 were regarded as mucoid and colors 3 to 5 as muco-purulent sputum therein after referred as purulent. The sputum specimens having reddish/rusty color was labeled as blood-stained sputum. Whenever the sputum specimens were heterogeneous, the predominant portion was considered to be the quality of sputum specimen.

Each sputum specimen was homogenized for smear preparation (Gram staining and AFB smear) culture of *M. tuberculosis* and Xpert® MTB/RIF assay (geneXpert).

### Microscopic assessment of specimen quality

Gram stained specimens were characterized according to modified Bartlett's screening criteria. Under the 10X objective; the average number of Neutrophils and Squamous Epithelial Cells (SEC) from three consecutive fields was recorded [12]. Sputum smear having an average number of <10 SEC and/or ≥25 Neutrophils / field was considered as good quality while smear with ≥10SEC and <25 Neutrophils/ field was unsatisfactory quality.

## Specimen processing and *Mycobacterium tuberculosis* detection

Zeihl Neelsen smear was done for all specimens and culture regardless of their Gram stain result. For culture, NALC-NaOH digestion-decontamination method was used according to the BACTEC™ MGIT™ 960 TB System protocol. Decontaminated samples were inoculated in Mycobacterium growth incubation tube (MGIT) and incubated in the MGIT™ 960 machine. Positive cultures were subjected to Ziehl-Neelsen (ZN) staining to confirm the presence of AFB. Further identification of *Mycobacterium tuberculosis* complex from the positive ZN cultures was done by use of immune-chromatographic analysis in this case Capilia TB assay (TAUNS Laboratories, Inc).

Sputum specimen processing for geneXpert was done according to manufacturer's recommendations. Briefly, the reagent buffer containing NaOH and isopropanol was added in a 2:1 ratio to at least 2 ml of the specimen. The mixture was incubated for 15 min with intermittent hand mixing. Two milliliters of the resulting liquefied inactivated sample was added into the Xpert® MTB/RIF cartridge (Cepheid, Sunnyvale CA, USA). The cartridge was placed in the instrument module, and the automated processes initiated. Results were automatically generated within 2 h and reported as MTB-negative or -positive (with semi-quantification) and rifampin (RIF) sensitive or resistant, error or invalid.

## Quality control

Internal quality control was performed throughout the sample processing process and key performance indicators were monitored. Briefly, at sample reception, an acceptance-rejection criterion was used in assessment of the samples received; artificial sputa was used alongside clinical specimens in the decontamination process; while positive and negative controls were used during staining process. New batch performance verification was done on media, identification kits and staining reagents before any new sets being used.

## Statistical analysis

We performed univariate analyses of participant characteristics, and bivariate analyses of culture positivity in relation to intrinsic and extrinsic factors. We defined intrinsic factors as those related to the subject and extrinsic factors as those related to the environment i.e. outside the body of the participants. The intrinsic factors analyzed included, clinical findings, comorbidity, socio-demographics, attitudes perceptions and beliefs. Extrinsic factors included financial considerations, and access to health care.

We performed all analyses using IBM SPSS, version 24.0 for Windows. Descriptive statistics such as mean (+/- standard deviation) were used to analyze continuous variables while frequencies and proportions were used to analyze categorical variables. Pearson's Chi-square was used at bivariate level to test for the association between culture positivity and different independent factors (intrinsic and extrinsic). Odds ratio (OR) with 95% CI was used to determine the magnitude/strength of the association. Binary logistic regression analysis was performed on the culture positivity using multiple intrinsic and extrinsic factors, identified to be significantly associated with culture positivity at bivariate level of analysis. Adjusted odds ratio (AOR) with 95% CI was used to determine the magnitude/strength of the association.

## Human subjects

The Kenya Medical Research Institute Scientific and Ethics Review Committee approved the study (Ref: KEMRI/SERU/CRDR/0013/3220). Written informed consent was obtained from all eligible study participants. This was witnessed by the study clinician.

## Results

### General characteristics of the participants

Data from a total of 202 participants aged 18 years and above were included; they consisted of 50.5% males and 49.5% females. A higher percentage (81.7%) of the participants were married, with some form of education where majority (52.5%) having primary level of education and 49.5% were self-employed. In regard to residence, 50.5% lived in urban areas (Table 1).

### Environmental and behavioral characteristics

Firewood was the preferred source of cooking fuel with 72.8% usage while liquid petroleum gas was least used 6.4%. Most participants (71.8%) lived over two kilometers from the tuberculosis testing facility. The motorcycle was the preferred means of transport (51.5%) to the hospital. Majority of the participants (75.7%) indicated they had never smoked. However, of those who smoked, (63.3%) had a history of more than 5years (Table 2).

### Knowledge of disease

Most participants were knowledgeable of tuberculosis disease, most had knowledge score category of greater than 51% (63.4%). From their responses, 119 (58.9%) indicated TB was caused by germs and was transmitted through coughing directly to others (57.4%). On symptoms

**Table 1. Demographic characteristics.**

| Variables | Total (n = 202) | |
|---|---|---|
| | n | % |
| **Gender** | | |
| Male | 102 | 50.5% |
| Female | 100 | 49.5% |
| **Age in years** | | |
| 18–30 | 66 | 32.7% |
| 31–45 | 69 | 34.2% |
| 46 and above | 67 | 33.2% |
| **Marital status** | | |
| Single | 49 | 24.3% |
| Married | 130 | 64.4% |
| Currently not married (divorced widowed or separated) | 23 | 11.4% |
| **Level of education** | | |
| No formal education | 37 | 18.3% |
| Primary | 106 | 52.5% |
| Secondary | 46 | 22.8% |
| Tertiary | 13 | 6.4% |
| **Occupation** | | |
| Formal employed | 21 | 10.4% |
| Self employed | 100 | 49.5% |
| Casual/others | 67 | 33.2% |
| Student | 14 | 6.9% |
| **Residence** | | |
| Urban | 102 | 50.5% |
| Sub-Urban | 41 | 20.3% |
| Rural | 59 | 29.2% |

**Table 2. Environmental and behavioral characteristics.**

| Variables | Total (n = 202) | |
|---|---|---|
| | n | % |
| **Energy** | | |
| Fire wood | 147 | 72.8% |
| Kerosene | 24 | 11.9% |
| Gas | 18 | 8.9% |
| Others | 13 | 6.4% |
| **Means of transport** | | |
| Bus/car | 36 | 17.8% |
| Motorcycle | 104 | 51.5% |
| Others | 62 | 30.7% |
| **Distance to the health facility** | | |
| >2KM | 145 | 71.8% |
| ≤2KM | 57 | 28.2% |
| **Smoking** | | |
| Current smoker | 20 | 9.9% |
| Former smoker | 29 | 14.4% |
| Never smoked | 153 | 75.7% |
| **History of smoking** | | |
| 1–5 years | 18 | 8.9% |
| Over 5 years | 31 | 15.3% |
| Never smoked | 153 | 75.7% |

related to TB, 81.7% mentioned it was characterized by a cough of 2 weeks, 79.2% night sweats, 66.8% weight loss and 53% chest pains. The chest X-ray (CXR) was the most popular (36.6%) diagnostic tool than sputum microscopy (5.9%) (Table 3).

## Perception, attitudes and TB treatment history

In this study, most of the participants (75.2%) visited the hospital more than 2weeks after symptoms appeared. Eighty (39.6%), 72 (35.6%, and 50(24.8%) had tuberculosis related symptoms for a duration of < 2weeks, 2–4weeks, and >4weeks, respectively. Twenty five (12.4%) of these participants had previous history of TB treatment of whom 8(32% were cured). The main reason documented for the delay in seeking treatment provided by 108 (71.1%) was that they neither felt very sick nor were they disturbed to seek for treatment (Table 4).

## Co-morbidities and concomitant medication

Of the enrolled participants 68(33.7%) indicated they suffered from other diseases other than the suspected tuberculosis. Of these, 48(70.6%) stated they had HIV and 46(65%) were on anti-retroviral therapy (ART). 11(5.4%) used herbal formulations (Table 5).

## Clinical signs and symptoms

Majority of the participants (98.5%) arrived in the hospital in a stable condition and walked unsupported. A higher proportion 117 (57.9%) of them had a normal body mass index (BMI) while 53(26.2%) were underweight and only 10(5%) were reported as obese. Conjunctive with pallor was seen in only 11(5.4%); while presence of lymphadenopathy was in 7(3.5%); oedema in 9(4.5%); hypertension was reported on 20 (9.9%) participants based on their systolic and

**Table 3. Knowledge of disease.**

| Variables | Total (n = 202) | |
|---|---|---|
| | n | % |
| **Cause of tuberculosis** | | |
| *Germs* | 119 | 58.9% |
| Hereditary | 20 | 9.9% |
| Witchcraft | 37 | 18.3% |
| Others | 26 | 12.9% |
| **Signs and symptoms** | | |
| *Fever* | 106 | 52.5% |
| *Cough 2 weeks* | 165 | 81.7% |
| *Night sweat* | 160 | 79.2% |
| *Weight loss* | 135 | 66.8% |
| *Chest pains* | 107 | 53.0% |
| **Knowledge on modes of TB transmission** | | |
| Sleeping in the same room with TB patient | 64 | 31.7% |
| *Patient coughing directly to others* | 116 | 57.4% |
| Sharing cups | 20 | 9.9% |
| Smoke | 12 | 5.9% |
| Dust | 14 | 6.9% |
| Heavy work | 7 | 3.5% |
| **Test TB diagnosis** | | |
| *CXR* | 72 | 35.6% |
| *Sputum microscopy* | 12 | 5.9% |
| Blood Culture | 22 | 10.9% |
| **Knowledge score categories** | | |
| <25% | 29 | 14.4% |
| 25–50% | 45 | 22.3% |
| 51–75% | 78 | 38.6% |
| >75% | 50 | 24.8% |

diastolic blood pressure; while 5(2.5%) abnormal heart sound, 101(50%) had reduced air entry and 107(53.3%) had difficulty in breathing (Table 6).

## Demographic factors associated with good quality specimen

Specimens were categorized as either good quality (Purulent/ Mucoid) or of unsatisfactory quality (salivary). From our findings, there was no association between gender, age, marital status, level of education or where people resided with production of good quality sputum. However formal employment (p = 0.009) as an occupation was associated with good quality sputum (Table 7).

## Environmental and behavioral characteristics associated with good quality specimen

There was no significant association on the type of cooking fuel; distance travelled to the health facility; or smoking history with the quality of sputum produced. However, the use of other means of transportation to the hospital other than motor cycles, bus/cars there was a significant association (p = 0.022) with good quality specimens (Table 8).

**Table 4. Perception, attitudes, practices and TB treatment history.**

| Variables | Total (n = 202) | |
|---|---|---|
| | n | % |
| **Time taken before visit the health facility after symptoms appeared** | | |
| <2 weeks | 50 | 24.8% |
| 2–4 weeks | 72 | 35.6% |
| >4 weeks | 80 | 39.6% |
| **If cough more than 2 weeks, reasons that may have caused the delay in visiting (n = 152)** | | |
| Fear of stigma | 9 | 5.9% |
| Fear of TB diagnosis | 10 | 6.6% |
| Distance from the health facility | 10 | 6.6% |
| Work | 6 | 3.9% |
| Not disturbed/Not very sick | 108 | 71.1% |
| Others | 9 | 5.9% |
| **Ever been treated for tuberculosis previously** | | |
| Yes | 25 | 12.4% |
| No | 177 | 87.6% |
| **If yes, the outcome? (n = 25)** | | |
| Cured | 8 | 32.0% |
| Failure | 3 | 12.0% |
| Out of control | 5 | 20.0% |
| Treatment completed | 6 | 24.0% |
| Unknown | 3 | 12.0% |

## Quality of specimen in relation to knowledge on TB perception, attitudes, practices, TB treatment history and co-morbidities and concomitant medication

There was a significant association (p<0.05) in production of good quality sputum and delay by more than 4 weeks to visit the hospital when any of TB symptoms first occurred as well as

**Table 5. Co-morbidities and concomitant medication.**

| Variables | Total (n = 202) | |
|---|---|---|
| | N | % |
| **Suffer from other disease** | | |
| Yes | 68 | 33.7% |
| No | 134 | 66.3% |
| **If yes, the disease (n = 68)** | | |
| Declined to respond | 20 | 29.4% |
| HIV | 48 | 70.6% |
| **On medication** | | |
| Yes | 70 | 34.7% |
| No | 132 | 65.3% |
| **If yes, the medication (n = 70)** | | |
| ART | 46 | 65.7% |
| Declined to respond | 24 | 34.3% |
| **On herbal treatment** | | |
| Yes | 11 | 5.4% |
| No | 191 | 94.6% |

**Table 6. Clinical signs and symptoms.**

| Variables | Total (n = 202) | |
|---|---|---|
| | N | % |
| **General condition** | | |
| Sick looking (requires support) | 3 | 1.5 |
| Stable (walking unsupported) | 199 | 98.5 |
| **BMI** | | |
| Underweight | 53 | 26.2 |
| Normal | 117 | 57.9 |
| Overweight | 22 | 10.9 |
| Obese | 10 | 5.0 |
| **Conjunctive** | | |
| Normal | 191 | 94.6 |
| Pallor | 11 | 5.4 |
| **Neck Lymphadenopathy** | | |
| No | 195 | 96.5 |
| Yes | 7 | 3.5 |
| **Hypertension** | | |
| Normal | 17 | 8.4 |
| Pre-hypertensive | 165 | 81.7 |
| Hypertensive | 20 | 9.9 |
| **Oedema** | | |
| Absent | 193 | 95.5 |
| Present | 9 | 4.5 |
| **Heart sounds** | | |
| Abnormal/ Added sounds | 5 | 2.5 |
| Normal | 197 | 97.5 |
| **Difficulty in breathing** | | |
| No | 95 | 47.0 |
| Yes | 107 | 53.0 |
| **Air entry** | | |
| Normal | 101 | 50.0 |
| Reduced | 101 | 50.0 |
| **Percussion note** | | |
| Dull | 98 | 48.5 |
| Resonant | 104 | 51.5 |
| **Auscultation** | | |
| Abnormal/added sounds | 100 | 49.5 |
| Normal breath sounds | 102 | 50.5 |

knowledge of TB. However, there was no association (p>0.05) with: when the sputum was produced spot at the hospital or as early morning at home; previous tuberculosis treatment; comorbidity; or use of herbal formulations (Table 9).

When we assessed association between good quality sputum production and clinical features including patients general condition, BMI, conjunctivitis, neck lymphadenopathy, hypertension, oedema, heart sounds, difficulty in breathing, air entry, percussion note and auscultation, an association (p = 0.047) was reported only with difficulty in breathing (Table 10).

**Table 7. Quality of specimen in relation to demographic characteristics.**

| Variables | Purulent/ Mucoid (n = 281) | | Salivary (n = 123) | | OR | 95% CI | | p value |
|---|---|---|---|---|---|---|---|---|
| | n | % | n | % | | Lower | Upper | |
| **Gender** | | | | | | | | |
| Male | 144 | 70.6% | 60 | 29.4% | 1.10 | 0.72 | 1.69 | 0.648 |
| Female | 137 | 68.5% | 63 | 31.5% | 1.00 | | | |
| **Age in years** | | | | | | | | |
| 18–30 | 86 | 65.2% | 46 | 34.8% | 1.00 | | | |
| 31–45 | 103 | 74.6% | 35 | 25.4% | 1.57 | 0.93 | 2.66 | 0.090 |
| 46 and above | 92 | 68.7% | 42 | 31.3% | 1.17 | 0.70 | 1.95 | 0.544 |
| **Marital status** | | | | | | | | |
| Single | 64 | 65.3% | 34 | 34.7% | 1.00 | | | |
| Married | 185 | 71.2% | 75 | 28.8% | 1.31 | 0.80 | 2.15 | 0.284 |
| Currently not married [A] | 32 | 69.6% | 14 | 30.4% | 1.21 | 0.57 | 2.58 | 0.613 |
| **Level of education** | | | | | | | | |
| No formal education | 49 | 66.2% | 25 | 33.8% | 1.00 | | | |
| Primary | 146 | 68.9% | 66 | 31.1% | 1.13 | 0.64 | 1.98 | 0.673 |
| Secondary | 66 | 71.7% | 26 | 28.3% | 1.30 | 0.67 | 2.51 | 0.444 |
| Tertiary | 20 | 76.9% | 6 | 23.1% | 1.70 | 0.61 | 4.77 | 0.313 |
| **Occupation** | | | | | | | | |
| Formal employed | 35 | 83.3% | 7 | 16.7% | 4.33 | 1.44 | 13.02 | **0.009** |
| Self employed | 139 | 69.5% | 61 | 30.5% | 1.98 | 0.89 | 4.40 | 0.096 |
| Casual/others | 92 | 68.7% | 42 | 31.3% | 1.90 | 0.83 | 4.34 | 0.129 |
| Student | 15 | 53.6% | 13 | 46.4% | 1.00 | | | |
| **Residence** | | | | | | | | |
| Urban | 145 | 71.1% | 59 | 28.9% | 1.26 | 0.78 | 2.05 | 0.351 |
| Sub-Urban | 58 | 70.7% | 24 | 29.3% | 1.24 | 0.67 | 2.28 | 0.490 |
| Rural | 78 | 66.1% | 40 | 33.9% | 1.00 | | | |

[A] This category includes divorcees, widows and those separated from their spouses.

Upon adjustment of confounders, the quality of sputum was good quality for patients who took more than 4 weeks to visit a hospital (p = 0.016; AOR 2.07; CI 11.5%-37.2%); patients that had conjunctivitis detected (p = 0.057; AOR 3.4; CI 9.6%-12%) and difficulty in breathing (p = 0.022; AOR 1.69; CI10.8%-26.5%). However the quality of sputum was adversely affected when one had some knowledge about the disease (p = 0.005; AOR 0.31; CI 14%-70%) this population had a tendency of producing salivary specimens (Table 11).

## Discussion

In this study we establish how various endogenous and exogenous factors linked to predisposition of an individual to tuberculosis affect quality of sputum produced by presumed tuberculosis patients. Of these, were well-established risk factors including; age, immunosuppression, nutrition status, comorbidity and emerging factors including, use of immunosuppressive drugs, indoor air pollution, alcohol, and tobacco smoking among others.

We evaluated clinical signs and symptoms associated with diagnosis of pulmonary tuberculosis (PTB) that influenced the quality of sputum. Difficulty in breathing, one of the respiratory symptom was associated with production of good quality sputum (p = 0.022; AOR 1.69; CI 95% 10.8–26.5). This symptom may have resulted from the effect of accumulation of pleural

**Table 8. Quality of specimen in relation to environmental and behavioral characteristics.**

| Variables | Purulent/ Mucoid (n = 281) | | Salivary (n = 123) | | OR | 95% CI | | p value |
|---|---|---|---|---|---|---|---|---|
| | n | % | n | % | | Lower | Upper | |
| **Fuel** | | | | | | | | |
| Fire wood | 203 | 69.0% | 91 | 31.0% | 1.39 | 0.61 | 3.19 | 0.431 |
| Kerosene | 36 | 75.0% | 12 | 25.0% | 1.88 | 0.67 | 5.23 | 0.229 |
| Gas | 26 | 72.2% | 10 | 27.8% | 1.63 | 0.55 | 4.76 | 0.376 |
| Others | 16 | 61.5% | 10 | 38.5% | 1.00 | | | |
| **Means of transport** | | | | | | | | |
| Motorcycle | 136 | 65.4% | 72 | 34.6% | 1.00 | | | |
| Bus/car | 49 | 68.1% | 23 | 31.9% | 1.13 | 0.64 | 2.00 | 0.680 |
| Others | 96 | 77.4% | 28 | 22.6% | 1.82 | 1.09 | 3.02 | **0.022** |
| **Distance to the health facility** | | | | | | | | |
| >2KM | 203 | 70.0% | 87 | 30.0% | 1.08 | 0.67 | 1.72 | 0.756 |
| ≤2KM | 78 | 68.4% | 36 | 31.6% | 1.00 | | | |
| **Smoking** | | | | | | | | |
| Current smoker | 30 | 75.0% | 10 | 25.0% | 1.37 | 0.64 | 2.92 | 0.412 |
| Former smoker | 41 | 70.7% | 17 | 29.3% | 1.10 | 0.60 | 2.04 | 0.756 |
| Never smoked | 210 | 68.6% | 96 | 31.4% | 1.00 | | | |
| **History of smoking** | | | | | | | | |
| 1–5 years | 24 | 66.7% | 12 | 33.3% | 0.91 | 0.44 | 1.90 | 0.811 |
| Over 5 years | 47 | 75.8% | 15 | 24.2% | 1.43 | 0.76 | 2.69 | 0.263 |
| N/A | 210 | 68.6% | 96 | 31.4% | 1.00 | | | |

effusion an exudate that usually has predominantly lymphocytes [13]. Several respiratory disease conditions including tuberculosis, pneumonia, chronic obstructive pulmonary disease (COPD), and lung cancer are associated with difficulties in breathing and may elicit similar symptoms. The production of good quality sputum in these cases is vital for definitive diagnosis.

During primary infection *Mycobacterium tuberculosis* multiplies in the lungs and causes mild inflammation while in the conjunctiva it may manifest with presence of ocular lesions [14–16]. This phenomenon is also shown with *Streptoccocus pneumonia* which can infect both the conjunctiva and cause pneumonia [17,18] other causes of conjunctivitis include immunologic factors [19] such as allergens and mechanical means [20]. Our findings show significant associations of inflamed conjunctiva and production of good quality sputum. Since pathogenic and non-pathogenic agents cause inflammation of both the conjunctiva and the lungs these agents can be ruled out quickly when good sputum is produced and advanced diagnostics are used during TB diagnosis.

Delays in TB diagnosis occur at both health system level and at patient level. Factors contributing to patient delay can be: socio-demographic factors such as the type of employment [21]. Specifically, we found that being on formal employment affected the quality of sputum produced (p = 0.009); this could be the reason for delays before seeking clinical attention which was significantly associated (p = 0.016) with production of good sputum quality. Delays before treatment are attributed to disease progression in the host. It has been recommended that TB diagnosis should be done within 21days after experiencing at least one tuberculosis symptoms [22, 23] but many studies show patients delay to seek health services more than a month from the onset of TB symptoms [24–28]. The paradox is that tuberculosis diagnosis delays are important in transmission dynamics of the disease, its control strategies and

**Table 9. Quality of specimen in relation to knowledge on TB, perception, attitudes, practices, TB treatment history and co-morbidities and concomitant medication.**

| Variables | Purulent/ Mucoid (n = 281) | | Salivary (n = 123) | | OR | 95% CI | | p value |
|---|---|---|---|---|---|---|---|---|
| | n | % | n | % | | Lower | Upper | |
| **Type of specimen** | | | | | | | | |
| Spot | 142 | 70.3% | 60 | 29.7% | 1.07 | 0.70 | 1.64 | 0.746 |
| Morning | 139 | 68.8% | 63 | 31.2% | 1.00 | | | |
| **Knowledge on TB** | | | | | | | | |
| <25% | 49 | 84.5% | 9 | 15.5% | 1.00 | | | |
| 25–50% | 64 | 71.1% | 26 | 28.9% | 0.45 | 0.19 | 1.05 | 0.065 |
| 51–75% | 99 | 63.5% | 57 | 36.5% | 0.32 | 0.15 | 0.70 | **0.004** |
| >75% | 69 | 69.0% | 31 | 31.0% | 0.41 | 0.18 | 0.94 | **0.034** |
| **Time taken before visit the health facility after symptoms appeared** | | | | | | | | |
| <2 weeks | 67 | 67.0% | 33 | 33.0% | 1.00 | | | |
| 2–4 weeks | 89 | 61.8% | 55 | 38.2% | 0.80 | 0.47 | 1.36 | 0.406 |
| >4 weeks | 125 | 78.1% | 35 | 21.9% | 1.76 | 1.01 | 3.08 | **0.048** |
| **Ever been treated for tuberculosis previously** | | | | | | | | |
| Yes | 37 | 74.0% | 13 | 26.0% | 1.28 | 0.66 | 2.51 | 0.466 |
| No | 244 | 68.9% | 110 | 31.1% | 1.00 | | | |
| **Suffer from other disease** | | | | | | | | |
| Yes | 93 | 68.4% | 43 | 31.6% | 0.92 | 0.59 | 1.44 | 0.715 |
| No | 188 | 70.1% | 80 | 29.9% | 1.00 | | | |
| **On medication** | | | | | | | | |
| Yes | 93 | 66.4% | 47 | 33.6% | 0.80 | 0.52 | 1.24 | 0.320 |
| No | 188 | 71.2% | 76 | 28.8% | 1.00 | | | |
| **On herbal treatment** | | | | | | | | |
| Yes | 18 | 81.8% | 4 | 18.2% | 2.04 | 0.67 | 6.15 | 0.199 |
| No | 263 | 68.8% | 119 | 31.2% | 1.00 | | | |

increased mortality of patients in the community [29] whereas accurate diagnosis is dependent on the quality of sputum produced.

Paramasivam *et al* in their study in India indicate that inadequate knowledge contributes to diagnosis delay [30] this augury contrasts with our study where by participants were knowledgeable but still delayed in seeking for treatment. Also, on health literacy- most of our participants knew signs and symptoms of tuberculosis (p = 0.005; AOR 0.31; 95% CI 14–70) but produced unsatisfactory specimens (salivary). The production of these specimens would further delay diagnosis with some diagnostic tools if used alone. The level of knowledge of the disease shows that TB advocacy in the community is bearing some fruits. However, even with this knowledge more should be done to motivate the health seeking behavior of patients. Nonetheless, distance from the health facility has been shown to cause delay in disease diagnosis. Upon adjustment of confounders, we found no association between the distance participants travelled and the quality of sputum produced. In our case participants travelled >2KM, most of them used motorized transportation including vehicles and motor cycles. Alternative modes of transport included walking and use of and boats. Adenager *et al*, study in Ethiopia showed that patients who travelled more than 2.5 Km were 1.6 times more likely to delay more than 21 days to contact a health facility than the ones who travelled less [31 This is also seen in studies done in Brazil, China and sub-Saharan African countries which reported factors including place of first consultation, travel time, or distance from the health facility to be associated with delays in TB diagnosis and treatment [25,31].

**Table 10. Quality of specimen in relation to clinical signs and symptoms.**

| Variables | Purulent/ Mucoid (n = 281) | | Salivary (n = 123) | | OR | 95% CI | | p value |
|---|---|---|---|---|---|---|---|---|
| | n | % | n | % | | Lower | Upper | |
| **General condition** | | | | | | | | |
| Sick looking (requires support) | 5 | 83.3% | 1 | 16.7% | 2.21 | 0.26 | 19.12 | 0.460 |
| Stable (walking unsupported) | 276 | 69.3% | 122 | 30.7% | 1.00 | | | |
| **BMI** | | | | | | | | |
| Underweight | 74 | 69.8% | 32 | 30.2% | 1.00 | | | |
| Normal | 162 | 69.2% | 72 | 30.8% | 0.97 | 0.59 | 1.60 | 0.914 |
| Overweight/ Obese | 45 | 70.3% | 19 | 29.7% | 1.02 | 0.52 | 2.02 | 0.945 |
| **Conjunctive** | | | | | | | | |
| Normal | 262 | 68.6% | 120 | 31.4% | 1.00 | | | |
| Pallor | 19 | 86.4% | 3 | 13.6% | 2.86 | 0.84 | 10.00 | 0.078 |
| **Neck Lymphadenopathy** | | | | | | | | |
| No | 272 | 69.7% | 118 | 30.3% | 1.00 | | | |
| Yes | 9 | 64.3% | 5 | 35.7% | 0.78 | 0.26 | 2.38 | 0.663 |
| **Hypertension** | | | | | | | | |
| Normal | 25 | 73.5% | 9 | 26.5% | 1.00 | | | |
| Pre-hypertensive | 226 | 68.5% | 104 | 31.5% | 0.78 | 0.35 | 1.74 | 0.546 |
| Hypertensive | 30 | 75.0% | 10 | 25.0% | 1.08 | 0.38 | 3.07 | 0.885 |
| **Oedema** | | | | | | | | |
| Absent | 267 | 69.2% | 119 | 30.8% | 1.00 | | | |
| Present | 14 | 77.8% | 4 | 22.2% | 1.56 | 0.50 | 4.76 | 0.438 |
| **Heart sounds** | | | | | | | | |
| Abnormal/ Added sounds | 9 | 90.0% | 1 | 10.0% | 4.04 | 0.51 | 32.22 | 0.155 |
| Normal | 272 | 69.0% | 122 | 31.0% | 1.00 | | | |
| **Difficulty in breathing** | | | | | | | | |
| No | 123 | 64.7% | 67 | 35.3% | 1.00 | | | |
| Yes | 158 | 73.8% | 56 | 26.2% | 1.54 | 1.00 | 2.35 | **0.047** |
| **Air entry** | | | | | | | | |
| Normal | 132 | 65.3% | 70 | 34.7% | 1.00 | | | |
| Reduced | 149 | 73.8% | 53 | 26.2% | 1.49 | 0.97 | 2.27 | 0.066 |
| **Percussion note** | | | | | | | | |
| Dull | 142 | 72.4% | 54 | 27.6% | 1.31 | 0.85 | 2.00 | 0.220 |
| Resonant | 139 | 66.8% | 69 | 33.2% | 1.00 | | | |
| **Auscultation** | | | | | | | | |
| Abnormal/added sounds | 146 | 73.0% | 54 | 27.0% | 1.38 | 0.90 | 2.12 | 0.136 |
| Normal breath sounds | 135 | 66.2% | 69 | 33.8% | 1.00 | | | |

We performed BMI calculation categorizing our participants as underweights, normal or obese to determine if there was any association with the quality of samples produced. There was no significant association by sputum quality. Nutritional status has also been assumed to have an obvious relationship with TB [32]. A review by Lönnroth *et al* 2010 shows a strong relationship between active TB and low BMI and this occurs across varying incidences of TB in different countries and across all levels of BMI [33]. We postulate that the strong association between TB and BMI especially for a low-BMI body build that predisposes one to TB [34] can be linked to the disease progression issue and not a detection problem due to sputum produced.

Human Immunodeficiency Virus (HIV) as a predisposisng factor for progression of TB [35] and intake of ART did not influence the sputum quality. Even though it is well established

**Table 11. Factors associated with good quality specimen (Purulent/Mucoid).**

| Variables | AOR | 95% CI | | p value |
|---|---|---|---|---|
| | | Lower | Upper | |
| **Knowledge on TB transmission** | | | | |
| <25% | 1.00 | | | |
| 25–50% | 0.43 | 0.18 | 1.03 | 0.057 |
| 51–75% | **0.31** | 0.14 | 0.70 | **0.005** |
| >75% | 0.49 | 0.21 | 1.15 | 0.102 |
| **Time taken before visit the health facility after symptoms appeared** | | | | |
| <2 weeks | 1.00 | | | |
| 2–4 weeks | 0.82 | 0.47 | 1.43 | 0.482 |
| >4 weeks | 2.07 | 1.15 | 3.72 | **0.016** |
| **Conjunctive** | | | | |
| No | 1.00 | | | |
| Yes | 3.40 | 0.96 | 12.05 | 0.057 |
| **Difficulty in breathing** | | | | |
| No | 1.00 | | | |
| Yes | 1.69 | 1.08 | 2.65 | **0.022** |

that TB diagnosis in HIV infected people especially those with a low CD4 count is complicated by lack of a productive cough [36] resulting to higher rates of sputum smear-negative disease. Our findings also showed no association between HIV patients on antiretroviral therapy (ART) and sputum quality. The use of ART may have normalized the pattern of disease to be more similar to that of HIV negative patients as also documented by [37].

Non-communicable diseases (NCD) and their contribution to TB progression and drug interactions during management are becoming important co-morbidities of study. Cough a clinical indicator for Lung cancer, other respiratory conditions of COPD and tuberculosis are associated with it. Lung cancer has similar clinical characteristic as tuberculosis including expectoration [38] to differentiate one from the other, clinical history and examination is important whereas sputum quality may be included as a marker for elevated suspicion for laboratory diagnosis. On the other hand, COPD is characterized by significant exposure to noxious particles or gases [39]. Patient history can is vital for diagnosis. In this study use of firewood a cause of indoor pollution and a risk factor of COPD [40] was not associated with quality sputum even though it was the source of cooking fuel for most of the participants. Our study also shows no association between sputum quality and hypertension even though most of the hypertensive cases presented good quality sputum specimens for TB diagnosis. There are however conflicting findings on tuberculosis and hypertension. Chung and colloquies show an association between TB and Hypertension [41], but five other studies reported no evidence to support an association between TB and hypertension between the control and hypertensive groups, [42–44].

## Conclusion

This study showed that both intrinsic and extrinsic factors affected the quality of sputum produced by presumed tuberculosis patients. Clinical and behavioral characteristics including conjunctivitis, difficulty in breathing and delay in seeking treatment were important factors that determined the production of good quality sputum specimens. It also showed that knowledge of tuberculosis disease does not translate to patients producing good quality sputum for diagnosis of the disease.

The TB program should also scale up health education to not only to improve TB awareness in the community but also to motivate presumed tuberculosis patients to produce specimens for accurate diagnosis.

## Supporting information

**S1 File. Intrinsic and extrinsic factors and sputum quality dataset final.** Complete dataset utilized for data analysis.
(CSV)

## Acknowledgments

We would like to thank the participants who participated in this study; the staff and administration of Malindi sub county Hospital; and the staff and the Director General Kenya Medical Research Institute Research for the support and permission to publish this article. We would also specifically like to thank Martha Njoroge, Evangeline Mathiu and Meryl Robi Chacha for their laboratory support.

## Author Contributions

**Conceptualization:** Fred Orina, Moses Mwangi, Evans Amukoye.

**Data curation:** Moses Mwangi.

**Formal analysis:** Moses Mwangi.

**Funding acquisition:** Fred Orina, Moses Mwangi.

**Investigation:** Fred Orina, Benson Kitole.

**Methodology:** Fred Orina, Moses Mwangi, Hellen Meme.

**Project administration:** Fred Orina, Moses Mwangi.

**Supervision:** Benson Kitole.

**Writing – original draft:** Fred Orina, Moses Mwangi, Hellen Meme, Evans Amukoye.

**Writing – review & editing:** Moses Mwangi, Hellen Meme, Evans Amukoye.

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
