## [Decision Letter · Decision Letter 0]

8 Oct 2019

PONE-D-19-25295

Intrinsic and extrinsic factors associated with sputum characteristics of presumed tuberculosis patients

PLOS ONE

Dear Dr. Orina,

Thank you for submitting your manuscript to PLOS ONE. After careful consideration, we feel that it has merit but does not fully meet PLOS ONE’s publication criteria as it currently stands. Therefore, we invite you to submit a revised version of the manuscript that addresses the points raised during the review process.

We would appreciate receiving your revised manuscript by Nov 22 2019 11:59PM. To enhance the reproducibility of your results, we recommend that if applicable you deposit your laboratory protocols in protocols.io, where a protocol can be assigned its own identifier (DOI) such that it can be cited independently in the future. For instructions see: http://journals.plos.org/plosone/s/submission-guidelines#loc-laboratory-protocols

We look forward to receiving your revised manuscript.

Kind regards,

HASNAIN SEYED EHTESHAM

Academic Editor

PLOS ONE

**Journal Requirements:**

2. Please provide additional details regarding participant consent. In the ethics statement in the Methods and online submission information, please ensure that you have specified (1) whether consent was informed and (2) what type you obtained (for instance, written or verbal, and if verbal, how it was documented and witnessed). If the need for consent was waived by the ethics committee, please include this information.

**Additional Editor Comments (if provided):**

Major Revision

**Comments to the Author**

1. Is the manuscript technically sound, and do the data support the conclusions?

Reviewer #1: Partly

Reviewer #2: No

2. Has the statistical analysis been performed appropriately and rigorously? 

Reviewer #1: Yes

Reviewer #2: No

3. Have the authors made all data underlying the findings in their manuscript fully available?

Reviewer #1: No

Reviewer #2: Yes

4. Is the manuscript presented in an intelligible fashion and written in standard English?

Reviewer #1: Yes

Reviewer #2: Yes

5. Review Comments to the Author

Reviewer #1: The manuscript does not fulfill the merit of the journal as it lacks novelty and provide any significant finding that is already not published.

Moreover several lacunae were identified regarding studying sputum characteristics, sample consistency analysis, mentioning references at appropriate places, type of culture method used, whether Direct Vs Indirect AFB staining process used, which ICT tests was used.

Due to major limitations in the manuscript it is found unsuitable for consideration.

Reviewer #2: Manuscript Number: PONE-D-19-25295

Manuscript Title: Intrinsic and extrinsic factors associated with sputum characteristics of presumed tuberculosis patients

Review:

1. There is discrepancy in explaining the extrinsic & intrinsic factors in the Statistical analysis section. The authors defined intrinsic factors as those related to the subject and extrinsic factors as those related to the environment (Outside the body of the subjects). Then why did the authors classify 'clinical findings' as extrinsic factors?

b) There is error in Table 5, " If yes, the medication (n=70)": the responses do not add up to 70.

c) The study deduces that "other means of transport" and "50-75% knowledge of TB transmission" among other factors responsible for good sputum production, but does not discuss the probable reasons due to which these factors effect the sputum production.

d) The paper is well written. However, the significance and outcome of the study is not clear and it does not provide any new information and knowledge.

6. PLOS authors have the option to publish the peer review history of their article (what does this mean?). If published, this will include your full peer review and any attached files.

Reviewer #1: No

Reviewer #2: No

---

## [Author Response · Author response to Decision Letter 0]

2 Dec 2019

Response to reviewers

Comments to the Author

1. Is the manuscript technically sound, and do the data support the conclusions?

Reviewer #1: Partly

Reviewer #2: No

Response: Thank you for this observation we have re-written the conclusion section to match the data presented

2. Has the statistical analysis been performed appropriately and rigorously? 

Reviewer #1: Yes

Reviewer #2: No

The statistical analysis was done from univariate analysis bivariate analysis and multivariate analysis. We think these can draw deduction on the factors that affect sputum production. We are open for any more suggestions and further discussions

3. Have the authors made all data underlying the findings in their manuscript fully available?

Reviewer #1: No

Reviewer #2: Yes

We wish to indicate that we have updated the data on supporting information.

4. Is the manuscript presented in an intelligible fashion and written in standard English?

Reviewer #1: Yes

Reviewer #2: Yes

 Response: Thank you for the observation

5. Review Comments to the Author

Reviewer #1: The manuscript does not fulfill the merit of the journal as it lacks novelty and provide any significant finding that is already not published.

Response: We wish to indicate that A few studies have been involved in elucidating the quality of sputum from tuberculosis patients through macroscopic and microscopic characterization while others have correlated the sputum quality and detection of tuberculosis using conventional methods or highly advanced molecular techniques. However, NO studies to the best of our knowledge have linked sputum quality and patient related factors or factors that predisposes individual to tuberculosis. We believe the information generated is new.

Moreover several lacunae were identified regarding studying sputum characteristics, sample consistency analysis, mentioning references at appropriate places, type of culture method used, whether Direct Vs Indirect AFB staining process used, which ICT tests was used.

Response: Than you for the review. We have update the methodology section and included the missing sections including the culture method used and quality control sections.

We however wish to clarify that we did not use direct or indirect AFB methods in this study since we used liquid culture for detection of cases. We believe culture is more sensitive that AFB staining methods in detection of TB cases.

We have also updated the methods section and indicate the use of capilia as the ICT used

Reviewer #2: Manuscript Number: PONE-D-19-25295

Manuscript Title: Intrinsic and extrinsic factors associated with sputum characteristics of presumed tuberculosis patients

Review:

1. There is discrepancy in explaining the extrinsic & intrinsic factors in the Statistical analysis section. The authors defined intrinsic factors as those related to the subject and extrinsic factors as those related to the environment (Outside the body of the subjects). Then why did the authors classify 'clinical findings' as extrinsic factors?

Response: Thank you for this observation we have revised and appropriately classified clinical findings as part of intrinsic factors.

b) There is error in Table 5, " If yes, the medication (n=70)": the responses do not add up to 70.

Response: Thank you for this observation, we have revised table 5 and responses add up.

c) The study deduces that "other means of transport" and "50-75% knowledge of TB transmission" among other factors responsible for good sputum production, but does not discuss the probable reasons due to which these factors effect the sputum production.

Response: We have reviewed the discussion section and deduced the probable reasons

d) The paper is well written. However, the significance and outcome of the study is not clear and it does not provide any new information and knowledge.

Thank you for the observation. We have reviewed our conclusions to show significance of the outcome. we believe to the best of our knowledge that we have linked sputum quality and patient related factors or factors that predisposes individual to tuberculosis. We believe the information generated is new.

---

## [Editor Report · Decision Letter 1]

13 Dec 2019

Intrinsic and extrinsic factors associated with sputum characteristics of presumed tuberculosis patients

PONE-D-19-25295R1

Dear Dr. Orina,

We are pleased to inform you that your manuscript has been judged scientifically suitable for publication and will be formally accepted for publication once it complies with all outstanding technical requirements.

With kind regards,

HASNAIN SEYED EHTESHAM

Academic Editor

PLOS ONE

Additional Editor Comments (optional):

I have gone through the Author’s response to reviewers comment and the revised manuscript. The Authors have revised the manuscript and appropriately classified clinical findings and these have now been included. Table 5 has been revised completely.

In summary, the revised version represents a novel study linking sputum quality and patient related factors to tuberculosis which have not been described so far. I recommend the revised version of this manuscript for publication.
---

## [Editor Report · Acceptance letter]

17 Dec 2019

PONE-D-19-25295R1 

Intrinsic and extrinsic factors associated with sputum characteristics of presumed tuberculosis patients 

Dear Dr. Orina:

I am pleased to inform you that your manuscript has been deemed suitable for publication in PLOS ONE. Congratulations! Your manuscript is now with our production department. 

With kind regards,

on behalf of

Prof HASNAIN SEYED EHTESHAM 

Academic Editor

PLOS ONE